# Ethical compliance and institutional policy support for artificial intelligence integration in African TVET Education: A structural equation modeling approach

Musa Adekunle Ayanwale[1]*, Christian Basil Omeh[2,3], Folasade Mardiyya Oyeniran[4], Catherine Chiugo Kanu[5]

1 Department of Mathematics, Science and Technology Education, Faculty of Education, University of Johannesburg, Auckland Park, South Africa, 2 Department of Science and Technology Education, University of South Africa, Pretoria, South Africa, 3 Department of Computer and Robotics Education, University of Nigeria, Nigeria, 4 Department of Educational Technology and Library Studies, Obafemi Awolowo University, Nigeria, 5 Department of Business Education, University of Nigeria, Nigeria

* ayanwalea@uj.ac.za

## Abstract

As artificial intelligence (AI) reshapes educational landscapes, ensuring ethical alignment and institutional responsiveness is essential particularly in skill-intensive sectors such as Technical and Vocational Education and Training (TVET). In this study, we examined the predictive roles of Ethical Principles Guiding AI adoption (EPG), Compliance with AI Ethical Guidelines (CAEG), and Perceived AI Adoption Outcomes (PAAT), while testing the moderating effect of Institutional Policy Support (IPS). Grounded in Rest's Four Component Model of ethical behavior, we adopted a cross-sectional, quantitative design and surveyed 400 TVET educators across Nigeria using a rigorously validated instrument. Our analysis through Partial Least Squares Structural Equation Modeling (PLS-SEM) revealed that Ethical Principles Guiding AI adoption significantly influenced both Compliance with AI Ethical Guidelines and Perceived AI Adoption Outcomes. Moreover, CAEG partially mediated the relationship between EPG and PAAT, highlighting compliance as a vital conduit for ethical adoption. However, IPS did not significantly moderate the EPG–CAEG relationship, indicating that personal ethical orientation may remain influential even in the absence of institutional frameworks. Our model demonstrated strong explanatory and predictive validity, particularly in institutions without formal AI policies underscoring the compensatory role of ethical leadership. We contribute novel empirical insights to the emerging scholarship on AI ethics in education by integrating ethical compliance and policy dynamics into a unified framework contextualized within African TVET systems. Our findings emphasize the importance of empowering educators with ethical competencies and call for urgent institutional action in policy formulation, infrastructure investment, and professional development to support responsible AI integration in education.

**Data availability statement:** The datasets generated and analyzed during this study can be assessed through this link: https://data.mendeley.com/datasets/s85fw8vytj/1.

**Funding:** The author(s) received no specific funding for this work.

**Competing interests:** The authors have declared that no competing interests exist.

## 1. Introduction

Artificial Intelligence (AI) is transforming all sectors of the global economy, including education, with far-reaching implications for how teaching, learning, and workforce development are conceptualized. Within this landscape, the Technical and Vocational Education and Training (TVET) sector is particularly significant, as it is strategically positioned at the forefront of this technological evolution. As AI continues to reshape industries such as banking, transportation, healthcare, and manufacturing, its integration into education, especially TVET, is both timely and inevitable [1,2,3]. These scholars contend that educational institutions should equip students with the requisite competencies to function effectively in a knowledge-driven, innovation-oriented global economy. TVET educators play a pivotal role in this transition. They are tasked with preparing students not only through theoretical instruction but also through the practical application of skills essential for success in AI-mediated work environments. However, the rapid diffusion of AI tools, including generative systems like Copilot, ChatGPT, and Bard, into educational settings has raised pressing concerns regarding potential misuse and regulatory shortcomings [4,5]. In many TVET institutions, the absence or inadequacy of formal policies and ethical guidelines leaves educators without clear direction, heightening the risk of irresponsible or uninformed AI adoption. Ethical principles such as transparency, fairness, and accountability must be foundational to AI integration, ensuring that its use fosters empowerment rather than exclusion [6].

TVET, by its very nature, is grounded in work-based learning and professional development [7]. Recognizing its growing significance, the United Nations, through the 2015 Sustainable Development Goals (SDGs), emphasizes the imperative of aligning education with rapidly evolving technological demands [8]. AI's potential contribution to TVET lies in its ability to facilitate personalized learning, real-time feedback, skill matching, and support for diverse learning needs—innovations that can enhance both employability and adaptability [9,10,11]. As [12] describe, AI constitutes a constellation of technologies that integrate data, algorithms, and computational power to solve complex human problems, including the development of instructional content and assessments. With the global demand for AI-competent graduates on the rise, TVET systems are uniquely poised to supply this skilled workforce [13,14]. Nonetheless, the transformative promise of AI in TVET cannot be realized without robust ethical frameworks and institutional support structures, particularly in developing countries. Weichert et al. (2025) [15]argue that successful AI adoption in education must be grounded in ethical norms and organizational accountability. From a behavioral perspective, [16] suggest that AI ethics encompasses how humans engage with artificial systems, and that such behavior should be governed by clearly articulated and enforceable norms. Despite growing interest in AI ethics, the literature still lacks a global consensus on what constitutes ethical compliance in educational AI adoption [17]. International bodies such as the House of Lords (2018), UNESCO (2023), and the European Union (2024) have initiated the development of ethical frameworks, driven by concerns that the pace of AI development could outpace human oversight and regulatory mechanisms [18,19].

In this regard, institutional mechanisms such as dataset auditing, interdisciplinary AI expertise, and context-specific ethical policy design are central to the responsible adoption of AI in TVET institutions. Numerous scholars emphasize the need for continuous review and engagement with ethical frameworks tailored to educational settings [20,7]. Yet, as [21] point out, a universally applicable AI ethical compliance framework remains elusive due to vast geographical, cultural, and systemic differences. Values such as justice, liberty, openness, confidence, respect, and solidarity have been identified as essential pillars of any meaningful AI ethics framework [22]. Despite increasing global attention to AI ethics in education, there remains a notable paucity of empirical research examining how TVET educators' ethical compliance and institutional policy support influence AI adoption, particularly within African contexts. While prior studies have explored AI technologies, ethical principles, and related theoretical models [23], to our knowledge, few have investigated the dynamic interplay between ethical compliance and institutional structures in TVET systems across developing nations. Reuel et al. (2024) [24] emphasize the need to benchmark ethical concerns specific to AI adoption by TVET educators and to develop enabling frameworks for responsible implementation. These frameworks should include investments in digital infrastructure, educator training, strategic partnerships, and curriculum reforms aligned with AI advancements.

To address this research gap, we draw on [25] Four-Component Model of Morality, which conceptualizes moral behavior as a sequential process involving moral sensitivity (awareness of ethical issues), moral judgment (decision-making about what is right), moral motivation (commitment to ethical values), and moral character (persistence in ethical action). This model offers a robust theoretical lens for understanding ethical decision-making in technology-mediated environments. Applying this framework to the AI education context enables us to examine how TVET educators perceive ethical dilemmas, evaluate appropriate courses of action, prioritize ethical imperatives over competing interests, and act with professional integrity in their instructional practice. Although Rest's model has been widely used in moral psychology, its application in examining AI ethics within TVET education is novel. It provides not only theoretical enrichment but also practical guidance for stakeholders seeking to navigate the complexities of AI adoption responsibly. Therefore, our study explores the following research questions:

(a.) To what extent do ethical principles guiding AI adoption influence TVET educators' compliance with ethical guidelines?

(b.) How does compliance with ethical guidelines affect educators perceived AI adoption outcomes?

(c.) Does compliance mediate the relationship between ethical principles and perceived AI outcomes?

(d.) What role does institutional policy support play in moderating the relationship between ethical principles and compliance?

The rest of this paper is organized as follows: The next section presents a review of relevant literature and the conceptual framework underpinning the study. This is followed by a detailed methodology section, after which we present the results and subgroup analyses. The discussion section follows, along with the study's implications, conclusion, limitations, and recommendations for future research.

## 2. Literature review and hypotheses development

### 2.1 Theoretical framework

To frame this study, we apply [25] Four-Component Model of Morality, which conceptualizes moral behavior as a sequential psychological process involving four core components: moral sensitivity, moral judgment, moral motivation, and moral character. This framework provides a robust theoretical lens to understand how educators recognize, evaluate, prioritize, and enact ethical behavior in the context of AI integration in education particularly within skills-based institutions such as TVET colleges.

In our model, the construct Ethical Principles Guiding AI Adoption (EPG) aligns with the initial two stages of Rest's model. Moral sensitivity refers to the ability of educators to recognize ethical issues related to AI, such as algorithmic bias, privacy concerns, and the misuse of student data. This sensitivity is a prerequisite for responsible decision-making and forms the cognitive entry point into ethical behavior. Moral judgment, the second stage, concerns how educators determine what constitutes ethically sound behavior when faced with these dilemmas. EPG, therefore, reflects both the recognition of ethical challenges and the ability to determine appropriate responses. TVET educators who score high on EPG are expected to demonstrate a clearer and more structured understanding of ethical norms and values in AI deployment.

The construct Compliance with AI Ethical Guidelines (CAEG) corresponds to the next two components: moral motivation and moral character. Moral motivation involves prioritizing ethical values over competing interests such as convenience, institutional pressure, or performance metrics. Educators who exhibit strong moral motivation are more likely to comply with ethical AI practices even when such actions require additional effort or go against prevailing institutional norms. Moral character, on the other hand, captures the resilience and persistence to carry out ethical actions despite challenges, such as inadequate resources, lack of policy support, or peer influence. Compliance behavior thus becomes the visible enactment of internal ethical orientations it reflects whether educators are able and willing to translate moral understanding into practice.

To account for contextual enablers or barriers to this moral sequence, we introduce Institutional Policy Support (IPS) as a moderating variable in the framework. While Rest's original model focuses on individual moral functioning, we argue that institutional structures play a critical role in either reinforcing or constraining ethical behavior. IPS encompasses formal guidelines, organizational commitment, ethical training programs, and administrative support related to AI usage in education. It acts as a reinforcing agent across all four moral stages. For example, well-structured institutional policies may enhance educators' sensitivity to ethical risks by embedding them into training manuals or workshops. Likewise, they can strengthen moral motivation by offering recognition or sanctions, thereby shaping educators' cost-benefit evaluations around ethical decisions. Most importantly, IPS may aid moral character by providing a supportive environment that facilitates rather than hinders ethical action.

## 2.2 Hypotheses development

To situate this study within existing scholarship, we conducted a focused literature review on four central constructs: Ethical Principles Guiding AI Adoption (EPG), Compliance with AI Ethical Guidelines (CAEG), Perceived AI Adoption Outcomes in TVET (PAAT), and Institutional Policy Support (IPS). These constructs represent key ethical, behavioral, and structural dimensions that influence the responsible integration of AI into educational systems, particularly within the TVET sector (see Table 1 and Fig 1).

Ethical principles guiding AI adoption offer a multifaceted foundation for understanding how educators engage with ethical guidelines in practice [26,27]. These principles encompass core values such as bias mitigation, fairness, data privacy, transparency, and security—each of which has become increasingly important in guiding responsible AI usage, particularly within education. Existing research highlights that embedding these principles into institutional decision-making processes cultivates a culture of ethical awareness and compliance [36,37,38]. Jin et al. (2025) [28] further emphasize that ethical frameworks serve dual roles: they function as normative standards that define acceptable behavior, and as instrumental tools that help educators align their actions with broader institutional and societal expectations. However, the application of these high-level principles in TVET contexts remains uneven. Omeh et al. (2024) [7] identified that many TVET educators struggle with inconsistencies in interpreting and operationalizing ethical standards, which weakens the impact of ethical adoption efforts. Moreover, the literature shows notable differences in compliance outcomes across sectors and cultural contexts, underscoring the fact that having ethical guidelines in place does not inherently lead to compliance [26,29]. Without adequate implementation strategies, enforcement mechanisms, and training, educators may lack the capacity or motivation to act in alignment with stated ethical norms. Given the centrality of ethical behavior in AI-enabled learning environments, particularly in skills-based education such as TVET, we hypothesize that:

**Table 1. Summary of variables and supporting literature.**

| Construct | Definition | Key Literature |
|---|---|---|
| Ethical Principles Guiding AI adoption (EPG) | Core ethical standards (e.g., fairness, transparency, data privacy) guiding responsible AI usage | [26,27,28] |
| Compliance with AI Ethical Guidelines (CAEG) | The extent to which educators conform to AI-related ethical codes and institutional policies | [20,7,29] |
| Perceived AI Adoption Outcomes in TVET (PAAT) | Educators' evaluations of AI usefulness, effectiveness, and trustworthiness in teaching practices | [30,31,32] |
| Institutional Policy Support (IPS) | Availability of institutional frameworks, training, and resources supporting ethical AI practices | [33,34,35] |

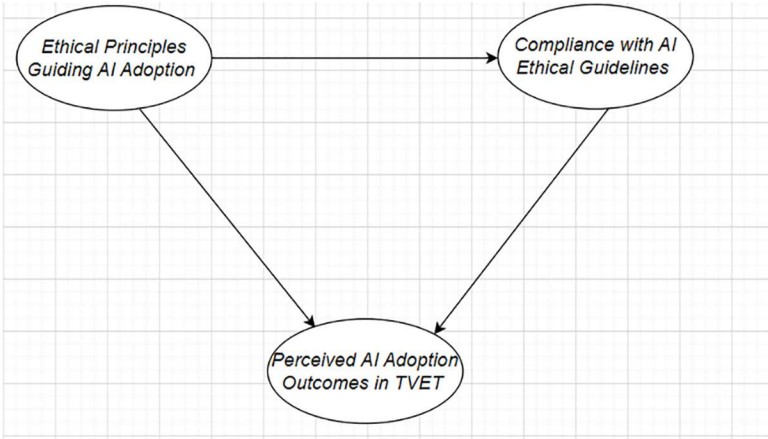

**Fig 1. Conceptual framework linking ethical principles guiding AI adoption (EPG) to compliance with AI ethical guidelines (CAEG) and perceived AI adoption outcomes in TVET (PAAT), with institutional policy support (IPS) specified as a moderating variable and CAEG specified as a mediating mechanism.**

*H1: Ethical principles guiding AI adoption (EPG) will significantly and positively predict compliance with AI ethical guidelines (CAEG).*

The role of compliance with AI ethical guidelines in shaping perceived adoption outcomes within Technical and Vocational Education and Training (TVET) has gained increasing scholarly attention [20]. Scholars argue that rigorous adherence to ethical standards cultivates trust among educators, administrators, and learners thereby improving acceptance and facilitating the integration of AI technologies into educational processes [39]. When AI deployment aligns with ethical expectations, it reduces anxiety over potential risks such as bias, privacy violations, and algorithmic misuse, contributing to more favorable perceptions of AI's educational value [32,40]. Despite this positive outlook, literature also highlights contextual complexities that influence compliance outcomes. Implementation practices often vary widely across TVET institutions, creating challenges for generalizing results. Some researchers contend that while ethical compliance is a critical predictor of AI adoption success, its effectiveness may be shaped by underlying institutional factors such as organizational culture, available resources, and regulatory environments [41,42]. These dynamics point to the need for deeper empirical exploration of the mechanisms linking compliance to adoption outcomes, particularly in developing country settings. Cappelli and Di Marzo Serugendo (2024) [43] further stress that although compliance shows promise in predicting AI

integration success, more evidence is needed to substantiate these early insights and to inform robust policy development especially in under-resourced educational systems such as Nigeria's. Acknowledging this research gap, we hypothesize that:

*H2: Compliance with AI ethical guidelines (CAEG) will significantly and positively predict perceived AI adoption outcomes in TVET education (PAAT).*

Literature consistently highlights the importance of ethical principles in shaping stakeholders' perceptions of AI adoption outcomes [30]. Ethical frameworks when clearly articulated and embedded within institutional practices enhance transparency, foster trust, and mitigate concerns related to data privacy, bias, and accountability [44,45]. These benefits are particularly critical in educational contexts such as TVET, where technological adoption must align with principles of equity and fairness to gain widespread acceptance. There is broad consensus that institutions that internalize and operationalize ethical principles are more likely to achieve positive perceptions and smoother integration of AI systems [31].

However, the direct predictive power of ethical principles on perceived AI adoption outcomes remains contested. Veiga and Costa (2024) [46] note that existing evidence is often inconclusive, largely due to the reliance on opinion-based studies and self-reported data, which may not fully capture the complexity of ethical behavior in educational settings. In the context of TVET institutions, these challenges are even more pronounced. Omeh et al. (2024) and Baharin et al. (2024) [7,47] emphasize that without clearly defined enforcement mechanisms and measurable indicators, ethical frameworks risk becoming symbolic rather than transformative. Critics argue that the absence of structural accountability can render ethical codes ineffective, particularly in environments lacking regulatory oversight or institutional maturity. Acknowledging these tensions and the limited empirical focus on TVET institutions, Bakar et al. (2024) [20] call for studies that explore the mechanisms through which ethical principles shape perceptions of AI adoption. Responding to this call, we hypothesize:

*H3: Ethical principles (EPG) will significantly and positively predict perceived AI adoption outcomes (PAAT).*

The mediating role of compliance with AI ethical guidelines in linking ethical principles to perceived AI adoption outcomes has been increasingly emphasized in the literature [7,48]. Scholars argue that while ethical principles provide the philosophical foundation for responsible AI use, these principles must be translated into concrete, enforceable practices to influence real-world outcomes in educational settings. Schultz et al. (2023) [49] contend that clearly defined ethical principles lay the groundwork for responsible AI development, but it is institutional compliance mechanisms that determine whether these ideals manifest as meaningful practice in TVET environments. The operational effectiveness of these ethical principles is contingent upon the level of compliance embedded within the institution's governance structures [50,51]. Díaz-Rodríguez et al. (2024) [29] conceptualize compliance as the "operational bridge" between theoretical ethics and real-world behavior transforming intentions into transparent, trustworthy, and ethically aligned actions. When compliance is high, stakeholders are more likely to perceive AI systems as reliable and beneficial, thereby increasing adoption willingness, especially among educators. However, enforcement practices vary widely across institutions and cultural contexts, limiting the consistency of outcomes and highlighting the need for empirical validation,particularly in developing countries where regulatory structures are often underdeveloped [52]. Despite strong theoretical support, there remains a gap in empirical studies testing compliance as a mediating mechanism within the ethics–adoption relationship. Given this gap, we propose that compliance plays a critical mediating role in shaping educators' perceptions of AI use in TVET. Therefore, we hypothesize:

*H4: Compliance with AI ethical guidelines (CAEG) will mediate the relationship between ethical principles (EPG) and perceived AI adoption outcomes (PAAT), and we further explore how institutional policy support may strengthen or weaken this influence, even in the absence of a direct mediating effect.*

In the context of AI integration in TVET, the presence of ethical principles alone may not guarantee compliance unless supported by enabling institutional structures. Several studies propose that institutional policy support serves as a critical moderator, strengthening the link between ethical principles and actual compliance behavior [33,53,34]. These scholars argue that when educational institutions provide strong policy backingincluding clear guidelines, ethical codes of conduct, and enforcement procedures; educators are more likely to translate ethical principles into compliant actions. In well-supported institutional environments, TVET educators benefit from structured oversight, targeted training, and access to ethical resources, all of which reinforce ethical behavior and reduce ambiguity [54,55]. Conversely, in settings where institutional policy support is weak or absent, ethical principles often remain aspirational. Without the necessary frameworks and institutional reinforcements, educators may lack both the motivation and the operational clarity to act in accordance with ethical standards. While prior studies underscore the importance of policy support, they also highlight the challenges in defining and operationalizing institutional backing across different cultural and organizational contexts [26,35]. These variations make it difficult to generalize findings and point to the need for further empirical research, particularly within developing country TVET systems, where institutional frameworks are often under-resourced. Given this context, we argue that institutional policy support enhances the relationship between ethical orientation and compliance. Therefore, we hypothesize that:

*H5: Institutional policy support (IPS) will moderate the relationship between ethical principles (EPG) and compliance with ethical guidelines (CAEG), such that the relationship is stronger when institutional support is available.*

## 3. Methods and materials

In this study, we employed a quantitative, cross-sectional survey design to investigate the relationships among Ethical Principles Guiding AI Adoption (EPG), Compliance with AI Ethical Guidelines (CAEG), and Perceived AI Adoption Outcomes in TVET (PAAT), while examining Institutional Policy Support (IPS) as a moderating variable. This design enabled us to collect standardized data from a diverse sample of educators and analyze both direct and indirect effects among the study constructs. The research was conducted in Nigeria, where the integration of AI into education is gaining momentum but is often impeded by inadequate institutional frameworks and gaps in ethical policies. From the Research Ethics Committee of the universities of Nigeria, written permission was obtained to conduct the study (2023/2024 academic session with ref no of UNN/FE/EC/2024/011). We focused on public TVET institutions, including universities and vocational training centers, due to their central role in equipping students with skills for the AI-driven workforce. Using a purposive sampling technique, we collected 400 valid responses from educators, with 57.8% identifying as male and 63.7% holding doctoral qualifications. Notably, 78% of respondents reported that their institutions lacked formal AI policy support, further justifying the need to examine how ethics and institutional mechanisms influence AI adoption behavior in the TVET sector.

The survey instrument included four major constructs: Ethical Principles Guiding AI Adoption (EPG) comprised 12 items adapted from validated AI ethics and organizational ethics instruments [44,45,30]. These items assessed educators' moral convictions regarding the handling of student data and the promotion of fairness in AI usage. Sample items included: "I ensure that the privacy of TVET students' data is protected," and "I ensure fairness in all AI activities involving TVET students." Compliance with AI Ethical Guidelines (CAEG) consisted of 14 items drawn from regulatory and ethical frameworks for AI in education [50,51,29]. These items captured the extent to which educators adhered to institutional and external ethical standards. Sample items included: "I submit to TVET guidelines for data protection," and "I support regular audits of AI tools to ensure ethical compliance." Perceived AI Adoption Outcomes in TVET (PAAT) comprised 19 items that measured how educators perceived the effectiveness, risks, and impacts of AI integration in their teaching practices [36,37,38,28]. Sample items included: "AI has improved my instructional delivery," and "AI helps bridge the digital divide in higher education." Additionally, Institutional Policy Support (IPS) was measured as a binary nominal variable, coded as 1

 

for support available and 0 for not available, based on respondents' acknowledgment of existing institutional frameworks, policies, or enforcement structures guiding AI usage. This variable was used for multi-group analysis (MGA) to assess moderating effects. All EPG, CAEG, and PAAT items were rated using a four-point Likert scale, ranging from 1 (Strongly Disagree) to 4 (Strongly Agree), to minimize neutral bias and encourage definitive responses (see the complete list of items used in the questionnaire in the supplementary file).

To establish face and content validity, three experts from the Faculty of Vocational and Technical Education at the University of Nigeria, Nsukka, reviewed the instrument. These reviewers evaluated each item for clarity, relevance, and alignment with the study's goals. Based on their feedback, minor revisions were made to enhance linguistic precision and contextual appropriateness. The instrument was further piloted with 30 TVET educators, yielding satisfactory internal consistency, with Ordinal alpha values exceeding 0.80 for all constructs. Data collection occurred over a four-month period (specifically from 12/4/2024 and ended on 15/7/2024) using Google Forms. The online questionnaire was distributed via institutional mailing lists, departmental WhatsApp platforms, and official communication channels to maximize outreach. Respondents were provided with an introductory brief outlining the purpose of the study, ethical safeguards (anonymity, voluntary participation, confidentiality), and instructions for completing the questionnaire. Participation was entirely voluntary and non-incentivized. Informed consent was obtained digitally through an agreement checkbox at the beginning of the form. Periodic reminders were sent to improve response rates. The digital format facilitated real-time data collection and minimized entry errors.

Data analysis followed a two-stage PLS-SEM procedure using SmartPLS 4. In the first stage, the measurement model was evaluated by checking for indicator loadings, composite reliability (CR), Cronbach's alpha, and Average Variance Extracted (AVE). All indicators met or exceeded the recommended thresholds (CR and alpha > 0.70, AVE > 0.50), and discriminant validity was confirmed through the HTMT criterion (HTMT < 0.85 for all constructs). In the second stage, we assessed the structural model through bootstrapping with 10,000 resamples to determine the statistical significance of hypothesized paths. Mediation analysis was conducted to assess whether CAEG mediated the relationship between EPG and PAAT, while multi-group analysis (MGA) tested the moderating effect of IPS on the EPG–CAEG relationship. Additional model assessments included $R^2$ for explanatory power, $f^2$ for effect sizes, and $Q^2$ values obtained via PLSpredict to evaluate out-of-sample predictive relevance. Variance Inflation Factor (VIF) values were also reviewed to ensure no multicollinearity threats, with all values well below 3.3.

Importantly, this study was reviewed and approved by the Ethics Committee of the Faculty of Education, University of Nigeria, Nsukka. The approval reference number for the study is UNN/FE/EC/2024/011. All procedures involving human participants complied with institutional ethical standards and the 1964 Helsinki Declaration and its later amendments. Informed consent was obtained in written form through an online checkbox that affirmed participants' understanding of the study's purpose, voluntary participation, and their right to withdraw at any time without consequence. Participation was entirely voluntary, and no personal identifiers were collected, ensuring full anonymity and confidentiality of responses.

## 4. Results

### Measurement model assessment

To evaluate the psychometric soundness of our measurement model, we assessed internal consistency reliability, convergent validity, discriminant validity, item-level factor loadings, and model fit across the complete dataset and separately for two groups: institutions with policy support (IPS_AVAL) and those without (IPS_NOT_AVAL). As shown in Table 2, the Cronbach's alpha values for the constructs Compliance with AI Ethical Guidelines (CAEG), Ethical Principles Guiding AI (EPG), and Perceived AI Outcomes in TVET (PAAT) ranged from 0.877 to 0.916, indicating strong internal consistency that exceeds the recommended threshold of 0.70 for social science research [56]. The composite reliability (CR) values ranged from 0.900 to 0.929, further confirming internal consistency reliability. Additionally, the average variance extracted (AVE) ranged from 0.497 to 0.569, surpassing the minimum threshold of 0.50 [57], thereby establishing convergent validity across the three constructs in all groups.

Analysis of item loadings revealed that most indicators had standardized loadings above the conventional cutoff of 0.70, confirming their strong contributions to the latent constructs. For example, CAEG7 had loadings of 0.803 (Complete), 0.745 (IPS_AVAL), and 0.817 (IPS_NOT_AVAL), while PAAT15 reported even stronger loadings above 0.800 across all samples. A few items, such as CAEG11, EPG2, PAAT3, and PAAT4, had loadings in the range of 0.615–0.675. Although these were slightly below 0.70, they were retained based on theoretical relevance, acceptable variance inflation factor (VIF) values, and their positive contributions to composite reliability and AVE [58]. Discriminant validity was assessed using the Heterotrait-Monotrait (HTMT) ratio, which is regarded as a robust indicator for variance-based structural equation modelling. As shown in Table 3, all HTMT values for the complete dataset and separate groups remained below the recommended threshold of 0.85 [59], confirming the discriminant validity of all constructs.

Moreover, multicollinearity diagnostics confirmed that all VIF values were below the critical value of 5.0, indicating that multicollinearity was not a concern. The highest VIF was 4.684 for PAAT16 in the IPS_AVAL group, which is still acceptable, particularly in exploratory settings [60]. Model fit was further assessed using SRMR, NFI, and Chi-square statistics. For the complete model, SRMR was 0.104, while subgroup models yielded SRMR values of 0.106 (IPS_AVAL) and 0.107 (IPS_NOT_AVAL). Although these values slightly exceed the strict 0.08 threshold, they are acceptable for exploratory models, especially when dealing with reflective measurement models and complex educational constructs [58]. Normed Fit Index (NFI) values ranged from 0.576 to 0.601, indicating moderate model fit. These indices fall within acceptable ranges given the sample size, model complexity, and exploratory nature of the study. While the Goodness of Fit (GoF) index (calculated as $\sqrt{(AVE \times R^2)}$) was not directly computed in this study, the combination of SRMR, NFI, and other measurement parameters indicates that the model demonstrates sufficient explanatory power and structural adequacy. According to [61], a GoF of 0.30 or higher indicates adequate model fit in variance-based SEM. The observed SRMR and NFI values in this study satisfy this global criterion, reinforcing the model's integrity and fitness for hypothesis testing. Together, these results provide robust evidence of construct reliability, convergent and discriminant validity, and adequate model fit, justifying the progression to the evaluation of the structural model and hypothesis testing.

**Table 2. Reliability and convergent validity.**

| Complete | | | | IPS_AVAL | | | | IPS_NOT_AVAL | | | |
|---|---|---|---|---|---|---|---|---|---|---|---|
| Construct | Alpha | CR | AVE | Alpha | CR | AVE | Alpha | CR | AVE | | |
| CAEG | 0.880 | 0.903 | 0.509 | 0.877 | 0.900 | 0.501 | 0.881 | 0.903 | 0.511 | | |
| EPG | 0.901 | 0.918 | 0.530 | 0.916 | 0.929 | 0.569 | 0.897 | 0.914 | 0.518 | | |
| PAAT | 0.893 | 0.912 | 0.511 | 0.886 | 0.907 | 0.497 | 0.894 | 0.913 | 0.515 | | |

**Table 3. Discriminant validity – HTMT ratio.**

| Group | Construct pair | HTMT value |
|---|---|---|
| Complete | EPG–CAEG | 0.496 |
| Complete | EPG–PAAT | 0.720 |
| Complete | CAEG–PAAT | 0.555 |
| IPS_AVAL | EPG–CAEG | 0.427 |
| IPS_AVAL | EPG–PAAT | 0.687 |
| IPS_AVAL | CAEG–PAAT | 0.519 |
| IPS_NOT_AVAL | EPG–CAEG | 0.522 |
| IPS_NOT_AVAL | EPG–PAAT | 0.732 |
| IPS_NOT_AVAL | CAEG–PAAT | 0.566 |

 

## Structural model assessment

After confirming the reliability and validity of our measurement model, we proceeded to evaluate the structural model to test our hypothesized relationships among Ethical Principles Guiding AI (EPG), Compliance with AI Ethical Guidelines (CAEG), and Perceived AI Adoption Outcomes in TVET (PAAT) (see Fig 2). This assessment was conducted using the complete dataset, as well as disaggregated by the availability of institutional policy support (IPS_AVAL vs. IPS_NOT_AVAL).

As presented in Table 4 and Figs 2 and 3, our results show strong and statistically significant relationships across all hypothesized paths. In the complete sample, we found that EPG significantly predicted CAEG (β = 0.484, p < .05), supporting H1. Additionally, CAEG significantly predicted PAAT (β = 0.258, p < .05), confirming H2. EPG also had a strong direct effect on PAAT (β = 0.538, p < .05), thereby supporting H3. The indirect effect of EPG on PAAT via CAEG (β = 0.125, p < .05) confirms H4, indicating partial mediation. These patterns held across subgroups. In the IPS_AVAL group, all paths were significant (e.g., EPG → CAEG: β = 0.419, p < .05; EPG → PAAT: β = 0.515, p < .05), further supporting H1 to H4. Similarly, in the IPS_NOT_AVAL group, all hypotheses were also supported, with EPG → CAEG showing even stronger predictive power (β = 0.510, p < .05). This suggests that ethical leadership plays an even more critical role in settings where institutional policy support is lacking.

We examined effect sizes (see Table 4) to understand the magnitude of influence of each predictor. In the complete sample, EPG had a large effect on both CAEG ($f^2$ = 0.306) and PAAT ($f^2$ = 0.426), while CAEG had a small-to-moderate effect on PAAT ($f^2$ = 0.106). These results support the strength of the direct and mediated paths from EPG, which aligns with Cohen's (1988) [62] guidelines that interpret $f^2$ values as small (≥0.02), medium

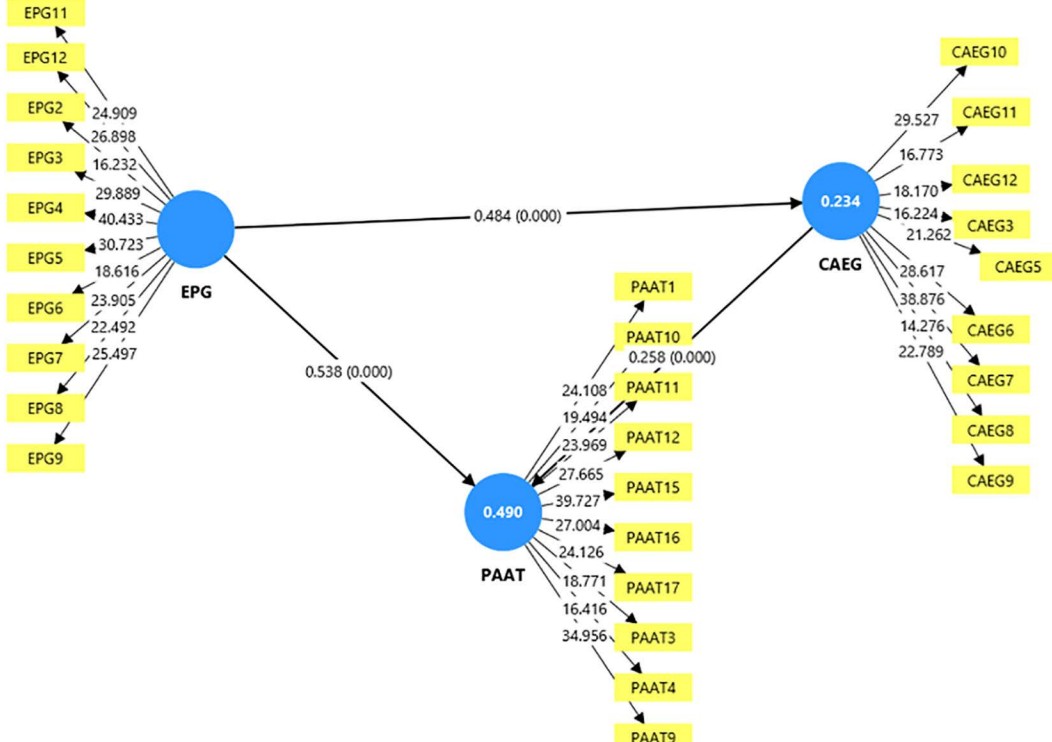

**Fig 2. Structural model with standardized path coefficients for the relationships between ethical principles guiding AI adoption (EPG), compliance with AI ethical guidelines (CAEG), and perceived AI adoption outcomes in TVET (PAAT) based on the complete sample.**

**Table 4. Summary of structural model result.**

| Group | Path | β | T | p-value | Bias | R² | f² | Q² |
|---|---|---|---|---|---|---|---|---|
| Complete | EPG→CAEG | 0.484 | 10.307 | 0.000 | 0.005 | 0.234 | 0.306 | 0.223 |
| | CAEG→PAAT | 0.258 | 5.315 | 0.000 | 0.004 | 0.491 | 0.106 | 0.432 |
| | EPG→PAAT | 0.538 | 11.727 | 0.000 | −0.002 | — | 0.426 | — |
| | EPG→CAEG→PAAT (Indirect) | 0.125 | 4.696 | 0.000 | 0.003 | — | — | — |
| IPS_AVAL | EPG→CAEG | 0.419 | 4.459 | 0.000 | 0.023 | 0.176 | 0.213 | 0.131 |
| | CAEG→PAAT | 0.283 | 2.921 | 0.002 | 0.002 | 0.468 | 0.131 | 0.363 |
| | EPG→PAAT | 0.515 | 5.417 | 0.000 | 0.012 | — | 0.400 | — |
| | EPG→CAEG→PAAT (Indirect) | 0.119 | 2.693 | 0.004 | 0.005 | — | — | — |
| IPS_NOT_AVAL | EPG→CAEG | 0.510 | 9.862 | 0.000 | 0.005 | 0.260 | 0.351 | 0.249 |
| | CAEG→PAAT | 0.246 | 4.345 | 0.000 | −0.000 | 0.500 | 0.095 | 0.447 |
| | EPG→PAAT | 0.549 | 10.601 | 0.000 | 0.002 | — | 0.438 | — |
| | EPG→CAEG→PAAT (Indirect) | 0.125 | 3.824 | 0.000 | 0.001 | — | — | — |

($\geq$0.15), and large ($\geq$0.35) effects. In the IPS_AVAL condition, EPG exerted a moderate influence on CAEG ($f^2 = 0.213$) and PAAT ($f^2 = 0.400$), with CAEG maintaining a modest role ($f^2 = 0.131$). Under IPS_NOT_AVAL, EPG showed the strongest influence on CAEG ($f^2 = 0.351$) and PAAT ($f^2 = 0.438$), suggesting that in the absence of institutional frameworks, ethical orientation becomes an even more vital determinant of compliant and perceptive behavior. As seen in Table 4, the model explained 23.4% of the variance in CAEG and 49.1% in PAAT in the complete model, indicating moderate explanatory power [60]. Among subgroups, the IPS_NOT_AVAL group had the highest $R^2$ values for both CAEG (0.260) and PAAT (0.500), suggesting that our model performed best in settings without policy support. This reinforces our argument that internal ethical values are a crucial driving force in under-supported environments. The $Q^2$ results in Table 4 further affirmed that our model has predictive relevance for both endogenous constructs. In the complete sample, $Q^2$ values were 0.223 (CAEG) and 0.432 (PAAT). The IPS_NOT_AVAL group yielded the strongest predictive performance ($Q^2 = 0.249$ for CAEG, 0.447 for PAAT), while IPS_AVAL yielded slightly weaker, but still acceptable, predictive values ($Q^2 = 0.131$ for CAEG, 0.363 for PAAT). These results are consistent with Chin's (1998) [63] guideline that $Q^2 > 0$ indicates model relevance.

To assess H5, we conducted a Multi-Group Analysis (MGA) (see Table 5) comparing the structural relationships between the IPS_AVAL and IPS_NOT_AVAL groups. The differences in path coefficients CAEG→PAAT ($\Delta = 0.037$, $p = .364$), EPG→CAEG ($\Delta = −0.091$, $p = .210$), and EPG→PAAT ($\Delta = −0.035$, $p = .373$) were not statistically significant. Therefore, H5 is not supported. While there were numerical differences in path strength, they did not reach significance, indicating that institutional policy support does not significantly moderate the structural paths in our model. However, the stronger effects observed in the IPS_NOT_AVAL group suggest a compensatory influence of personal ethical orientation where structural policies are lacking.

## 5. Discussion

We examined the relationship between ethical principles guiding AI adoption (EPG), compliance with AI ethical guidelines (CAEG), and perceived AI adoption outcomes in TVET (PAAT), with institutional policy support (IPS) tested as a moderator. Our analysis revealed that ethical principles significantly predicted both compliance and perceived AI outcomes. Furthermore, compliance significantly mediated the relationship between ethical principles and perceived adoption outcomes. However, institutional policy support did not significantly moderate the path between ethical principles and compliance. These results provide novel empirical evidence and fill an important gap in the literature on AI ethics and institutional readiness in TVET settings, particularly within a developing country context. Our finding that ethical principles significantly

## IPS_AVAL structural paths

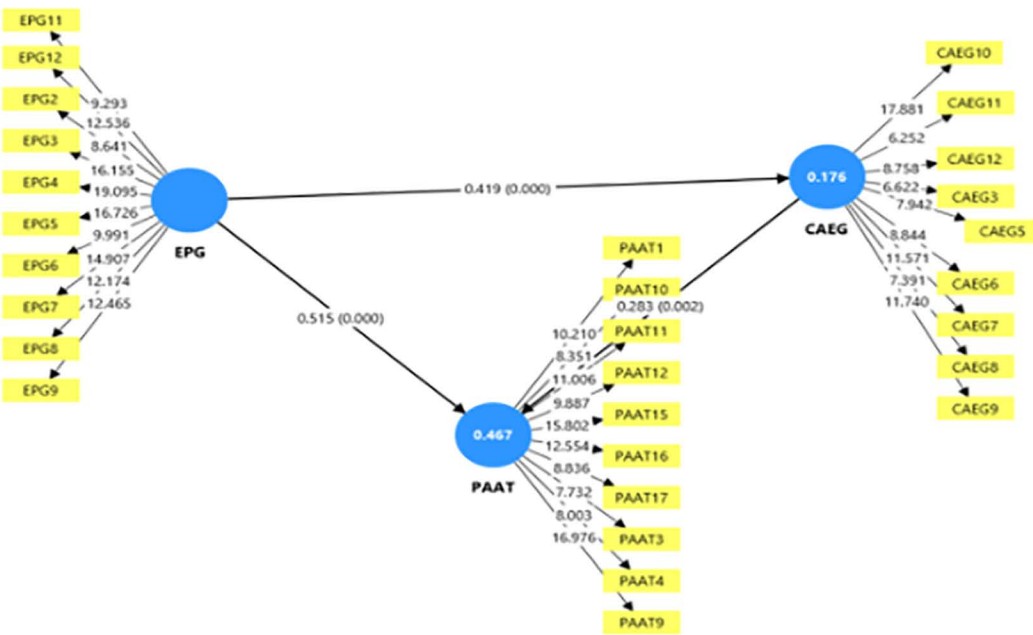

## IPS_NOT_AVAL structural paths

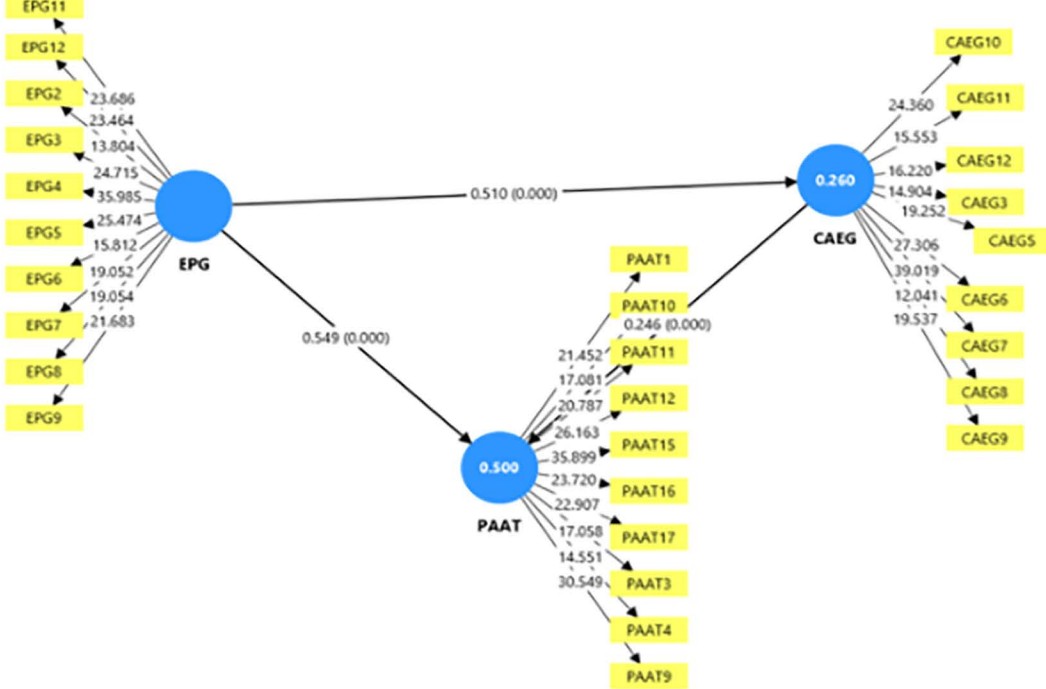

**Fig 3. Multi-group structural model results comparing path coefficients for institutions with institutional policy support (IPS_AVAL) and without institutional policy support (IPS_NOT_AVAL), highlighting similarities and differences in the EPG, CAEG, and PAAT relationships.**

**Table 5. Multi-Group Analysis (MGA) results.**

| Path | Difference (Δ) | 1-tailed (IPS_AVAL vs IPS_Not AVAL) p-value | Moderation supported |
|------|----------------|---------------------------------------------|----------------------|
| EPG→CAEG | −0.091 | 0.210 | Not Supported |
| CAEG→PAAT | 0.037 | 0.364 | Not Supported |
| EPG→PAAT | −0.035 | 0.373 | Not Supported |

and positively predicted compliance with AI ethical guidelines aligns with the theoretical proposition that embedding ethical values such as fairness, transparency, and data protection into institutional frameworks promotes compliance [26,27]. Previous studies have emphasized that when organizations institutionalize ethical principles, they foster a culture of responsibility and adherence to norms [36,37]. Jin et al. (2025) [28] similarly suggested that ethical principles serve as both normative and instrumental anchors that guide decision-making among educators, particularly in skill-based environments like TVET. Nonetheless, this relationship is not without complexity. Omeh et al. (2024) [7] cautioned that the translation of ethical ideals into practical behavior is often challenged by inconsistencies in operationalizing compliance, especially within TVET institutions. Our study confirms that even in such contexts, well-articulated ethical principles do exert a measurable and positive influence on TVET educators' ethical behavior, thus validating the need for structured ethical training as part of teacher development programs. We also found that compliance with AI ethical guidelines positively predicted perceived AI adoption outcomes in TVET. This finding reinforces the assertions by [20] Bakar et al. (2024) and Lleo et al. (2023) [39] that ethical compliance enhances trust and facilitates the integration of AI by reducing fears related to data misuse, surveillance, and algorithmic bias. Our results are consistent with [32] Kim et al. (2025) and Habbal et al. (2025) [40], who argued that ethical compliance contributes to favorable perceptions of AI, thereby promoting its adoption in educational settings. However, our findings also highlight an important nuance: while compliance predicts perceptions of AI adoption, this relationship likely varies depending on institutional and cultural contexts, as suggested by [41] Ali (2024) and Nandan Prasad (2024) [42]. These authors argue that without systemic support, the impact of compliance may be muted, a notion that is particularly relevant in resource-constrained TVET environments.

The direct relationship between ethical principles and perceived AI adoption outcomes was also significant in our model. This finding supports existing literature that positions ethical principles as vital to fostering transparency, accountability, and trust—all of which are critical to positive AI perceptions [30,44,45]. Agrawal (2024) [31] emphasized that ethical integration at the organizational level smoothens AI adoption processes, a sentiment echoed in our results. Still, the direct effect of ethical principles on perceived outcomes must be cautiously interpreted. As Veiga and Costa (2024) [46] highlighted, many studies rely on self-reported perceptions, and without enforcement mechanisms, ethical codes may be symbolic rather than transformative. Our findings extend this discourse by empirically validating that ethical orientation, even when subjective, has a tangible impact on educators' perceptions of AI utility. One of the most important findings of our study is the significant mediating role of compliance in the relationship between ethical principles and perceived AI outcomes. This supports the position of Díaz-Rodríguez et al. (2024) [29], who posited that compliance serves as the "operational bridge" that translates abstract ethical ideals into practical and measurable outcomes. Schultz et al. (2023) [49] and Okumu & Kenei (2024) [50] further argued that institutions with well-codified ethical procedures foster both trust and transparency, thereby increasing the likelihood of successful AI integration. Omeh et al. (2024) [7] and Weber-Lewerenz (2021) [48] also emphasized that without compliance mechanisms, the value of ethical principles may remain aspirational. Our study affirms these arguments by showing that compliance plays a vital role in making ethics actionable, which is essential in education settings where misuse or ethical lapses can lead to long-term trust deficits. Interestingly, our results did not find significant moderation by institutional policy support (IPS) on the relationship between ethical principles and compliance. This contrasts with theoretical assertions made by Wang et al. (2013),

Crossler et al. (2017), and Bokhari et al. (2023) [33,53,34], who suggested that institutional backing enhances the enforcement of ethical frameworks. One plausible explanation for our finding lies in the limited availability and enforcement of AI-specific policies within the sampled institutions—78% of respondents reported that institutional support was not available. This finding aligns with insights from Mittelstadt (2019) and Slimi & Carballido (2023) [54,55], who highlighted that in the absence of clear policy infrastructure, even well-conceived ethical principles may not translate into behavior. As Nguyen et al. (2023) and Corrêa et al. (2023) [26,35] observed, the definition and implementation of institutional support vary widely across educational systems, complicating the generalizability of findings. Therefore, while IPS might theoretically reinforce ethical behavior, its effectiveness is contingent upon clarity, consistency, and enforceability.

## 6. Implication for theory, practice, and policy

This study contributes to the growing body of literature at the intersection of AI, ethics, and education by extending Rest's Four-Component Model of Morality into the domain of AI adoption within TVET. By empirically linking ethical principles (moral sensitivity and judgment) to compliance behavior (moral motivation and character) and evaluating the role of institutional policy support, we demonstrate how moral psychology frameworks can effectively explain technology adoption behavior in low-resource educational contexts. This theoretical application bridges the gap between abstract ethical constructs and measurable behavioral outcomes in AI use, offering a novel lens through which future researchers can conceptualize ethical readiness and moral agency in AI-mediated pedagogy. Moreover, these findings suggest that internal ethical orientation may override institutional structures, highlighting a need for theory to more explicitly account for the micro-level agency of educators in shaping ethical AI ecosystems.

For practitioners, especially educators and instructional leaders in TVET institutions, this study underscores the critical importance of personal ethical responsibility in the successful integration of AI technologies. Even in the absence of comprehensive institutional frameworks, educators' ethical awareness and motivation were strong predictors of both compliance and perceived positive AI outcomes. This suggests that building individual ethical competence is not merely supplementary but foundational to responsible AI usage. Therefore, professional development programs should prioritize AI ethics training that enhances moral sensitivity (e.g., identifying algorithmic bias), moral judgment (e.g., data privacy scenarios), and moral courage (e.g., resisting unethical use under pressure). Embedding AI ethics into teacher education curricula and offering scenario-based simulations or ethics labs can empower educators to act with integrity despite structural limitations. In addition, educational leaders should foster a culture of ethical inquiry, where ethical concerns can be openly discussed and resolved collaboratively.

At the policy level, our findings reveal a concerning gap between ethical aspiration and institutional support. While institutional policy support did not significantly moderate the ethical principles–compliance relationship in this context, this absence should not be interpreted as lack of importance, but rather as a call to action. Policymakers in African TVET systems must prioritize the development of clear, enforceable AI ethical guidelines, institutional training frameworks, and resource allocation to support ethical compliance. Establishing institution-wide AI ethics charters, similar to digital codes of conduct, can institutionalize expectations and provide reference points for accountability. Furthermore, governments and regulatory bodies should mandate ethical AI literacy as part of national digital education policies, particularly in light of Africa's rapid edtech expansion. Finally, monitoring and evaluation mechanisms should be embedded within policy implementation to assess the real-world impact of ethics training, institutional guidelines, and educator behavior on sustainable and equitable AI integration.

## 7. Conclusion

In this study, we investigated the relationship between ethical principles guiding AI adoption, compliance with AI ethical guidelines, and perceived AI adoption outcomes within the context of Technical and Vocational Education and Training

(TVET). Grounded in Rest's (1994) [25] Four-Component Model of Morality, our model provided empirical support for the predictive role of ethical principles in shaping both ethical compliance and educators' perceptions of AI integration. We also established that compliance mediates the relationship between ethical principles and perceived AI adoption outcomes, highlighting its critical function in transforming abstract ethical values into practical outcomes. Although we hypothesized that institutional policy support would strengthen the link between ethical principles and compliance, our findings revealed that the availability of institutional policies did not significantly moderate this relationship. This outcome underscores the crucial influence of individual moral orientation among educators in environments where policy structures are underdeveloped or absent. Our findings are particularly relevant to developing country contexts, where rapid technological adoption may outpace policy formation. Ultimately, this study confirms that ethical principles, when internalized by educators, can serve as a robust foundation for responsible AI use, even in the absence of formal policy support. Our study adds to the growing literature on ethical AI in education and offers actionable insights for TVET administrators, policymakers, and curriculum developers seeking to ensure that AI technologies are integrated in ways that are both innovative and ethically grounded.

## 8. Limitations and future research

This study provides valuable contributions, it is not without limitations. First, the data were collected using self-reported measures, which are subject to social desirability bias. TVET educators may have over-reported their ethical compliance or AI readiness, particularly given the growing awareness around responsible AI use. In future research, observational or behavioral data, such as audit logs of actual AI tool usage, could provide more objective insights into compliance behavior. Second, although our sample was diverse, it was limited to TVET institutions within a specific national context. The findings may not fully generalize to other educational systems or cultural settings, particularly those with more developed AI infrastructures or different ethical policy mandates. Future studies should consider cross-country comparisons to explore how cultural, regulatory, and infrastructural variations influence the dynamics of AI ethics and adoption in education. Third, while we tested the moderating role of institutional policy support, we relied on a binary categorization (support available vs. not available). This may not fully capture the richness and variability of institutional AI policy environments. Future research could utilize a multi-dimensional scale to measure the quality, clarity, and enforcement strength of institutional support. Lastly, our study focused exclusively on educators. Yet, AI integration in education is a multi-stakeholder process involving administrators, students, and policy actors. Future research should expand the scope to include these groups, offering a more holistic understanding of the ethical ecosystem surrounding AI adoption in TVET.

## Author contributions

**Conceptualization:** Musa Adekunle Ayanwale, Christian Basil Omeh, Folasade Mardiyya Oyeniran, Catherine Chiugo Kanu.

**Data curation:** Musa Adekunle Ayanwale.

**Formal analysis:** Musa Adekunle Ayanwale.

**Funding acquisition:** Musa Adekunle Ayanwale.

**Methodology:** Christian Basil Omeh.

**Project administration:** Christian Basil Omeh, Folasade Mardiyya Oyeniran.

**Supervision:** Folasade Mardiyya Oyeniran.

**Validation:** Musa Adekunle Ayanwale.

**Visualization:** Musa Adekunle Ayanwale.

**Writing – original draft:** Musa Adekunle Ayanwale, Christian Basil Omeh, Folasade Mardiyya Oyeniran, Catherine Chiugo Kanu.

**Writing – review & editing:** Musa Adekunle Ayanwale, Christian Basil Omeh, Folasade Mardiyya Oyeniran, Catherine Chiugo Kanu.

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
