## [Decision Letter · Decision Letter 0]

26 Jul 2025

Dear Dr. Ayanwale,

Thank you for submitting your manuscript to PLOS ONE. After careful consideration, we feel that it has merit but does not fully meet PLOS ONE’s publication criteria as it currently stands. Therefore, we invite you to submit a revised version of the manuscript that addresses the points raised during the review process.

We look forward to receiving your revised manuscript.

Kind regards,

Bersissa Kumsa, DVM, MSc, PhD

Academic Editor

PLOS ONE

Journal Requirements:

3. In the online submission form, you indicated that the data underlying the results presented in the study are available from the corresponding author upon reasonable request.

4. Please ensure that you refer to Figure 1 in your text as, if accepted, production will need this reference to link the reader to the figure.

5. Please remove all personal information, ensure that the data shared are in accordance with participant consent, and re-upload a fully anonymized data set.

Additional Editor Comments :

Dear Authors,

The reviewers have completed their evaluation of your manuscript. I encourage you to revise and resubmit your work, ensuring that all reviewer comments are thoroughly addressed. Please incorporate the feedback carefully and provide a detailed, point-by-point response that clearly outlines every change made in response to the reviewers’ suggestions.

In addition, kindly correct all typographical and grammatical errors, and ensure that the manuscript is prepared in full compliance with the journal’s formatting and submission guidelines.

We look forward to receiving your revised submission.

Kind Regards,

Reviewers' comments:

Reviewer's Responses to Questions

**Comments to the Author**

1. Is the manuscript technically sound, and do the data support the conclusions?

Reviewer #1: No

Reviewer #2: Yes

2. Has the statistical analysis been performed appropriately and rigorously?

Reviewer #1: Yes

Reviewer #2: Yes

3. Have the authors made all data underlying the findings in their manuscript fully available?

Reviewer #1: Yes

Reviewer #2: Yes

4. Is the manuscript presented in an intelligible fashion and written in standard English?

Reviewer #1: Yes

Reviewer #2: No

Reviewer #1: Thanks for submitting your article to PLOS ONE journal. The topic titled “Ethical Compliance and Institutional Policy Support for Artificial Intelligence Integration in African TVET Education: A Structural Equation Modeling Approach” this topic is timely and contribute to the body of knowledge. The introduction and literature review was well articulated with robust analysis and good discussion section. Below are some observations to strengthen the draft to meet international readership

Introduction section

1. Break up long paragraphs for better readability and academic flow.

2. Avoid redundancy: phrases like “reshape sectors” and “technological transformation” are used repeatedly.

3. Ensure precision: clarify ambiguous phrases such as “Copilot, ChatGPT, and Bard” are these examples of large language models, generative tools, or AI platforms?

Some example of academic writing tone. Kindly review the draft to identify them and correct accordingly.

1. “AI is increasingly penetrating all sectors of the global economy…”

“AI is increasingly transforming sectors of the global economy…”

2. “finds itself at the forefront…”

“is positioned at the forefront…”

3. “must equip learners with the competencies…”

“should equip learners with the requisite competencies…”

4. “AI tools such as Copilot, ChatGPT, and Bard…”

“AI tools—including generative systems like Copilot, ChatGPT, and Bard…”

5. “ethical misuse and regulatory gaps”

“potential misuse and regulatory shortcomings”

6. “cannot be realized without strong ethical frameworks…”

“depends upon the establishment of robust ethical frameworks

7. “surpass human control and comprehension”

“outpace human oversight and understanding”

8. The IPS moderation result is discussed at length, which is good—but consider softening the tone slightly to account for sample-specific limitations:

“While IPS was not statistically significant in this context, its theoretical relevance remains compelling, particularly where robust policy infrastructure exists.

9. “Artificial Intelligence is reshaping global economies and educational paradigms, challenging traditional approaches to teaching and learning.” see this Consider integrating transitional phrases: For example,

“In light of this transformation…” or “Against this backdrop…” to signal a shift in focus.

Improve on the scope of the study was poorly articulated: There is need to contextualize the scope as to clearly show were the study was carried out and why. Clarify the scope of your study earlier. Perhaps add: “This paper focuses on the ethical integration of AI within TVET systems in developing countries, with an emphasis on institutional policy and educator behavior.”

Suggested Improvements

Clarity and Flow. Consider breaking longer sentences into shorter, more digestible units. For example:

1. “Ethical principles guiding AI adoption offer a multifaceted foundation for understanding how educators engage with ethical guidelines in practice” could be revised to: “Ethical principles provide a multifaceted foundation. They help explain how educators engage with guidelines in day-to-day practice.”

2. Phrases like “must be translated into concrete, enforceable practices to influence real-world outcomes” could be made more concise: "must be operationalized through enforceable practices to impact educational outcomes."

3. Avoid repeating similar ideas in adjacent sentences (e.g., the notion of institutional mechanisms transforming ideals into practice is echoed multiple times).

4. Use transitions to better link paragraphs. Phrases like “Building on this foundation…” or “This leads to…” help orient the reader.

5. The use of acronyms (EPG, CAEG, PAAT) is fine for brevity, but it may help to occasionally restate the full terms for readability—especially at the start of each hypothesis.

Strengthen Empirical Gap. The final paragraph acknowledges a lack of empirical studies in TVET settings, especially in developing countries. You could amplify this by highlighting why Nigeria presents a particularly compelling case (e.g., rapid digital expansion, policy vacuums, educator training gaps).

Clarify Theoretical Framing; You may want to clarify whether your proposed model is grounded in a specific theory (e.g., institutional theory, stakeholder theory, or ethics of care) to strengthen academic framing.

Methodology section

Clear Research Design, A cross-sectional survey design is aptly justified and well-aligned with the objectives of examining relationships among latent constructs. Contextual Relevance, You situate the study effectively within Nigeria’s emerging AI-in-education ecosystem, highlighting local institutional challenges. Instrument Robustness, Multiple measures (face/content validity, expert review, pilot testing) strengthen your instrument’s credibility. Using validated scales from recent literature boosts construct validity. Sampling Transparency, The demographics are clearly presented, with informative stats on gender and qualification levels. PLS-SEM Rationale and Execution

Discussion Section

Area of strength

1. Insightful Findings Interpretation: The triangulation of direct, mediating, and moderating effects is well-executed. You offer layered insight into how ethical principles translate into practice and shape perceptions of AI adoption.

2. Evidence-Based Argumentation: Your extensive engagement with relevant literature reinforces theoretical alignment and empirical relevance.

3. Real-World Relevance: The connection to Nigeria’s TVET ecosystem and resource constraints provides a compelling backdrop for the findings and implications.

4. Theoretical Contribution: Introducing Rest’s Four-Component Model adds originality. It also grounds your results in a moral psychology framework—a fresh angle in AI ethics literature.

5. Action-Oriented Implications: The dual strategy (bottom-up ethical awareness and top-down policy reforms) offers clear, feasible policy recommendations for institutions in developing countries.

Clear separate this part to enhance readability and flow of thought

1. Interpretation of Findings

2. Theoretical Implications

3. Policy Implications

4. Practical Implications

Avoid starting sentences with “Ultimately” or “While…” too often. They work well occasionally, but varying sentence openers improves rhythm.

Example tweak: “Ultimately, our study confirms…” “This study confirms…”

You might vary tone by swapping formal phrasing occasionally: “Our model provided empirical support…” “The findings offer empirical support…”

3. Expand Practical Implications Briefly: Consider adding one line on how curriculum developers or educators might act on these insights—for example, integrating AI ethics into training modules.

Reviewer #2: First of all, I am sure that this is an interesting topic, so it aroused my interest in reading it carefully. In the process, I found some problems that need to be addressed by the authors in order to publish with higher quality.

1. The abstract is well written.

2. The keywords can be adjusted. They don’t have to come from the title. You can design African education, digital education, and AI education.

3. Please check the literature I provided, because I have some doubts when writing the research gap. Please compare with these literatures and see what your contribution is.

4. The third and fourth research questions need to be revised. We are more concerned about how to influence rather than whether to influence. In addition, what if there is no mediating effect of institutional policies?

5. The introduction has too much content, delete half of it.

6. The theoretical basis and its application should be explained in the literature review.

7. Each variable should be explained in the literature review, preferably using a table.

8. The model diagram needs to be redrawn. This model diagram does not look academic enough. Please refer to other papers in the journal with model diagrams for adjustment.

9. After the results, we should draw the model graph we verified, indicating the regression coefficients and significance between the paths.

10. The discussion and conclusion are very good.

**Do you want your identity to be public for this peer review?** For information about this choice, including consent withdrawal, please see our Privacy Policy

Reviewer #1: No

Reviewer #2: No

---

## [Author Response · Author response to Decision Letter 1]

11 Sep 2025

The response to the reviewers' comments has been uploaded.

---

## [Editor Report · Decision Letter 1]

15 Oct 2025

Ethical Compliance and Institutional Policy Support for Artificial Intelligence Integration in African TVET Education: A Structural Equation Modeling Approach

PONE-D-25-32681R1

Dear Dr. Ayanwale,

We’re pleased to inform you that your manuscript has been judged scientifically suitable for publication and will be formally accepted for publication once it meets all outstanding technical requirements.

Kind regards,

Bersissa Kumsa, DVM, MSc, PhD

Academic Editor

PLOS ONE
---

## [Editor Report · Acceptance letter]

PONE-D-25-32681R1

PLOS One

Dear Dr. Ayanwale,

I'm pleased to inform you that your manuscript has been deemed suitable for publication in PLOS One. Congratulations! Your manuscript is now being handed over to our production team.

Kind regards,

on behalf of

Professor Bersissa Kumsa

Academic Editor

PLOS One